# Extended Lymphadenectomy for Proximal Transverse Colon Cancer: Is There a Place for Standardization?

**DOI:** 10.3390/medicina58050596

**Published:** 2022-04-26

**Authors:** Răzvan Cătălin Popescu, Florin Botea, Eugen Dumitru, Laura Mazilu, Luminița Gențiana Micu, Cristina Tocia, Andrei Dumitru, Adina Croitoru, Nicoleta Leopa

**Affiliations:** 1Department of General Surgery, Emergency Hospital of Constanța, 900591 Constanta, Romania; razvanpop2000@yahoo.com (R.C.P.); gherghe_nicoleta02@yahoo.com (N.L.); 2Faculty of Medicine and Pharmacy Constanta, Ovidius University, 900470 Constanta, Romania; eugen.dumitru@yahoo.com (E.D.); lauragrigorov@gmail.com (L.M.); citosan_plus@yahoo.com (L.G.M.); cristina.tocia@yahoo.com (C.T.); dr.andreidumitru@gmail.com (A.D.); 3“Dan Setlacec” Center of General Surgery and Liver Transplantation, Fundeni Clinical Institute, 022328 Bucharest, Romania; 4Faculty of Medicine, Titu Maiorescu University, 031593 Bucharest, Romania; adina.croitoru09@yahoo.com; 5Department of Gastroenterology, Emergency Hospital of Constanța, 900591 Constanta, Romania; 6Department of Oncology, Emergency Hospital of Constanța, 900591 Constanta, Romania; 7Department of Pathology, Emergency Hospital of Constanța, 900591 Constanta, Romania; 8Department of Medical Oncology, Fundeni Clinical Institute, 022328 Bucharest, Romania

**Keywords:** colon cancer, transverse colon, lymph node metastases, gastrocolic ligament, complete mesocolon excision

## Abstract

*Background and Objectives*: Complete mesocolon excision and high vascular ligation have become a standard procedure in the treatment of colon cancer. The transverse colon has certain embryological and anatomical particularities which require special attention in case of oncological surgeries. Proximal transverse colon cancer (TCC) can metastasize to the lymph nodes in the gastrocolic ligament. The aim of this study is to assess the tumor involvement of these lymph nodes and to determine the applicability of gastrocolic ligament lymph nodes dissection as the standard approach for proximal transverse colon cancer. *Materials and Methods*: this study analyzes the cases of patients admitted to the Surgery Department, diagnosed with proximal transverse colon cancer, with tumor invasion ≥ T2 and for which complete mesocolon excision with high vascular ligation and lymphadenectomy of the gastrocolic ligament (No. 204, 206, 214v) were performed. *Results*: A total of 43 cases operated during 2015–2020 were included in the study. The median total number of retrieved central lymph nodes was 23 (range, 12–38), that had tumor involvement in 22 cases (51.2%). Gastrocolic ligament tumor involvement was found in 5 cases (11.6%). The median operation time was 180 min, while the median blood loss was 115 mL (range 0–210). The median time of hospitalization was 6 days (range, 5–11). Grade IIIA in the Clavien-Dindo classification was noticed in 3 patients, with no mortality. Upon Kaplan–Meier analysis, tumors > T3 (*p* < 0.016) and lymph node ratio < 0.05 (*p* < 0.025) were statistically significant. *Conclusions*: lymph node dissection of the gastrocolic ligament in patients with advanced proximal transverse colon cancer may improve the oncological outcome in T3/T4 tumors, and therefore standardization could be feasible

## 1. Introduction

Colon cancer is one of the most common cancers, arousing particular interest to develop specific chemotherapy protocols and improve surgical techniques, as well as to increase survival by reducing the recurrence rate [1,2]. Despite significant progress in chemotherapy, surgery with radical intent remains the main curative treatment in colon cancer [3,4]. Complete mesocolon excision (CME) and central vascular ligation (CVL) have recently become standard surgical techniques and have significantly improved oncological results [4,5]. 

Particularly, the transverse mesocolon has embryological and anatomical features that make the dissection planes different from the embryological layers at this level. The relationship between the transverse colon, the great omentum, and the pancreas causes a fusion between these embryonic areas. Consequently, as demonstrated by Perrakis et al., tumors affecting the transverse colon spread beyond the latter’s embryological area [6]. In this sense, it has been suggested that lymphadenectomy for advanced proximal transverse colon cancer (TCC) should include excision of the gastrocolic ligament (GCL) [7]. The GCL is attached to the great gastric curvature, the first part of the duodenum and the transverse colon, merging posteriorly with the mesocolon (Figure 1), containing the gastroepiploic LN (No. 204), the infrapyloric LN (No. 206) and superficial pancreatic LN (No. 214v) [8]. The aim of this study is to report the frequency of tumoral involvement of these LN and to determine the role of the GCL lymph nodes (GCLN) dissection as standard approach for proximal TCC.

## 2. Materials and Methods

### 2.1. Study Group

A total of 43 patients who underwent surgery for proximal TCC between 2015–2020, by the same surgical team, were included in the study and were submitted to GCLN dissection. We defined the proximal transverse colon as the right half of the transverse colon, near the hepatic flexure. The inclusion criteria were: (1) proximal TCC; (2) depth of tumor invasion T2 to T4a; (3) CVL + CME + GCLN dissection; (4) histopathological analysis of more than 12 LN. The exclusion criteria were: (1) non-elective case; (2) cases with locoregional invasion/metastases in other organs; (3) any cancer personal history; (4) neoadjuvant therapy. Patients in the T1 stage were not included in the study because it was considered that a cancer in this stage does not have locoregional tumor involvement or metastases at the level of other organs, and the excision of the GCL cannot be taken into account [8,9].

Age-adjusted Charlson comorbidity index (ACCI) score was used to predict the risk of mortality and was calculated from a weighted index consisting of age and the number and seriousness of comorbid diseases. The Clavien–Dindo classification was used to categorize postoperative complications [10,11]. Major complications were considered at least grade IIIA complications. Postoperative mortality was defined as death within 90 days after surgery.

### 2.2. Surgical Technique

All the operations were undertaken under elective conditions, by the same surgical team. The cases that met the inclusion criteria were those of the proximal transverse colon. All patients underwent extensive right colectomy, from the terminal ileum to the distal transverse colon, including the ileocolic, right colic, middle colic vessels and gastrocolic trunk, according to the principles of CVL and CME. Patients with T3/4 tumors usually underwent open surgery, and those with T2 underwent laparoscopic surgery. Regardless of the surgery selected, it began with vessel approach, dissection of the ileocolic vessels and of the right side of the superior mesenteric artery. The resection length of the transverse colon and GCL was established to be approximately 10 cm distal to the tumor. GCL was detached from the great curvature of the stomach, the infrapyloric region and the anterior face of the pancreas, performing *en bloc* resection, including the right gastroepiploic vessels and LN stations No 204, 206 and 214v. The anastomosis was performed manually in all cases, regardless of the surgical approach. All patients received two drainage tubes. The specimen was marked with threads of different colors that highlighted the gastroepiploic artery, the gastroepiploic, infrapyloric and superficial pancreatic LN and further sent for pathological examination (Figure 2).

### 2.3. Study Endpoints

The primary endpoint was to determine the distribution of tumor invasion of the studied nodal stations (204, 206 and 214v), assessed by categorizing the LN retrieved from resected surgical specimens.

The secondary endpoints were to analyze the postoperative complications and to perform a comparison between the open and laparoscopic approaches.

### 2.4. Pathology

Serial cross-section at the level of specimen was performed, at intervals of 3–5 mm. All the LN obtained were formalin-fixed and H&E stained and were identified by experienced pathologists. (Figure 3).

The pathological data included: TNM classification, tumor staging, perineural, venous and lymphatic invasion. No. 204 LN were identified along the gastroepiploic artery, those in No. 206 along the pyloric artery to the confluence of the right gastroepiploic vein and those in No. 214 in the superficial pancreatic head and were assessed separately.

The lymph node ratio (LNR) was defined as the number of positive LN divided by the total number of LN harvested; patients were then divided into three LNR groups according to the following quartiles: LNR_0_ (<0.05), LNR_1_ (0.05–0.20) and LNR_2_ (>0.20). 

### 2.5. Follow-Up

Patients were followed every 3 months during the first postoperative year, every 6 months during the second year, and annually thereafter. Complete blood count, serum tumor marker, abdominal–pelvic and thoracic computed tomography, colonoscopy were performed in accordance with the oncological follow-up. 

Adjuvant chemotherapy was given based on the pathological report of each patient. No patients with stage I disease underwent adjuvant chemotherapy. Adjuvant chemotherapy was recommended for high-risk stage II and stage III–IV patients (using either 5-FU/leucovorin/capecitabine or 5-FU/oxaliplatin/leucovorin/capecitabine (FOLFOX or CAPOX)).

### 2.6. Statistical Analysis

For statistical analysis, (Statistical Package for Social Sciences) SPSS version 18 (IBM Corp.; Armonk, NY, USA) was used. Results are presented as mean ± standard deviation or medians with range. The categorical variables were expressed as count (percentage), and chi-square tests were used to compare demographic factors, as well as clinical–pathological parameters. Mean overall survival (OS) was estimated using the Kaplan–Meier test. OS was defined as the time between surgery and death, and patients alive at the last follow-up or lost during follow-up were censored. Independent risk factors for GCLN tumor involvement were determined using uni- and multivariate binary logistic regression models using positive GCLN as dependent variable. Odds ratios and 95% CIs were estimated. A *p* value of <0.05 was considered an indicator of statistical significance.

## 3. Results

The clinical and pathological characteristics of patients are depicted in Table 1. The mean age was 65.09 ± 12.63 years (range: 35–86). In total, 22 patients (51.2%) had an ACCI (Adjusted Age-Adjusted Charlson Comorbidity Index) score of 4–5 and 32.5% had an ACCI score ≥6. The median BMI was 26.3. A total of 30 patients (69.8%) underwent open surgery and 13 (30.2%) underwent laparoscopic surgery. The mean operative time was 193.14 ± 22.15 min (range, 150–240 min). The average blood loss was 114.19 mL ± 35.87 (range, 0–210 mL). 

The major complication rate was 7% (N = 3). All these patients had Dindo–Clavien grade IIIA complications: intraperitoneal collection requiring transparietal drainage (N = 1), pleural effusion requiring thoracentesis (N = 1), and anastomosis hemorrhage requiring endoscopic hemostasis (N = 1). Minor complication rate was 23.3% (Table 2): Dindo–Clavien grade II in 18.6% (N = 8), and grade I in 4.7% of cases (N = 2); wound site infections (N = 2); postoperative lymphatic leakage (N = 2) (treated conservatively), delayed gastric emptying (N = 6) (treated by medication and nasogastric tube). Postoperative mortality rate was nil.

The median postoperative hospital stay was 6 days (range, 5–11 days). After a median follow-up of 40.77 months, 10 patients (23.26%) died. The mean overall survival was 43.36 ± 20.45 months (range, 10.27–78.93). Three patients (7%) developed distant metastases (to the liver) and no patients developed local recurrence.

Most patients had stage III cancer (39.5%), followed by 34.9% with stage II cancer, and 14% with stage I cancer (Table 3). The mean number of dissected LN was 23 (range: 12–38). LN positivity was detected in 22 patients (51.2%). The mean number of involved LN was 2.95 (range: 0–16) for all patients, and 6.35 (range: 1–16) for LN-positive patients. LN positivity in the GCL area (11.6%) was observed in gastroepiploic region in three patients and in one patient, positive LN were identified in the gastroepiploic and infrapyloric LN. One patient (2.3%) had LN involvement in No. 214v and 204. This group of patients was considered to have tumor involvement in the extramesocolic area, being classified as UICC stage IVA. All these patients also presented with pericolic LN involvement. The mean LN ratio was 0.21 in patients with nodal positive status, and 0.37 in those with GCLN involvement.

A univariate and multivariate analysis for GCLN tumor involvement was performed (Table 4). Among the various parameters studied, T3/T4 (*p* = 0.019), lymphatic invasion (*p* = 0.011), venous invasion (*p* = 0.024) and perineural invasion (*p* = 0.008) were risk factors for GCLN tumor involvement found significant at the univariate analysis. Multivariate analysis revealed that the extramesocolic LN metastasis of the GCL was found to be associated with pT4 (*p* = 0.016), and lymphatic invasion (*p*< 0.001).

A comparison of clinical and operative outcomes between laparoscopic and open surgery for T3 proximal TCC are presented in Table 5. Only T3 tumors were considered for this comparative analysis, as T4 tumors were exclusively operated by open approach, while T2 tumors were almost always operated on using laparoscopy. Patients who underwent laparoscopic surgery were younger (*p* = 0.193) and had better ACCI scores (*p* = 0.163); patients’ BMIs did not differ significantly (*p* = 0.481). The operation time was significantly shorter in the open group (*p* = 0.007). The mean hospital stay for laparoscopic group was reduced compared with the open approach group (*p* = 0.026). No postoperative complications occurred in the laparoscopic group.

The overall survival rates in patients with proximal TCC according to Kaplan–Meier curve is analyzed in Figure 4. Survival gradually declined with the increase in depth of infiltration of the primary tumor, number of positive LN estimated, and status of distant metastases (Table 3, Figure 4). In terms of depth of tumor invasion, the overall survival shows a decrease in patients with at least pT3 (*p* = 0.016) (Figure 4A). OS in patients with positive GCLN was similar compared to the negative ones (*p* = 0.008); note that the GCLN positive group had only five patients (Figure 4B). Survival among the groups of LNR was significantly different (*p* < 0.025), in favor of LNR0 (Figure 4C). 

## 4. Discussion

In cases with advanced colorectal tumors, it was clearly proven that excision of as many LN as possible is important at least for estimating survival and determining the indication for adjuvant cancer treatment [12]. In patients with colorectal cancer, successful resection of metastatic foci, such as those located in the lung or liver, is associated with significant long-term survival [13], and, as a consequence, the resection of non-regional LN metastases in high risk patients [14] should be associated with an OS benefit. This is particularly significant in our country, as most colorectal tumors are still diagnosed in advanced stages, leading to a death count of 34/100,000 (recorded in 2016–2018), almost twofold higher than the average European range [15]. An appropriate treatment and a thorough follow-up plan should be given full consideration for colon cancer patients. To improve the survival rate and the prognosis of the disease, the treatment should be adapted to each patient, according to their clinical characteristics, risk factors, and tumor staging. 

Surgical accuracy, along with oncological treatment, can influence the long-term survival rates. The accuracy of postoperative staging also includes an accurate analysis of the LN of the specimen. LNR is a powerful factor to assess the prognosis after surgery in colon cancer patients, that should be combined with other factors for accurate assessment and optimal therapeutical strategy [16,17]. 

Lymphatic drainage of TCC is more intricate than initially considered, and extramesocolic metastases demonstrated in other studies support this idea [8,18,19]. In our clinical practice prior to this study, in patients with proximal TCC, we observed GCLN suspicious of tumor involvement which were dissected, marked, and sent to pathology. The discovery of positive GCLN led us to include the dissection of these LN in the standard CME and CVL technique, in cases with at least T2 proximal TCC. In our study, GCLN involvement were detected in five patients (11.6%) with T2 or deeper invasive proximal TCC; all the patients with metastasis presented with LN positivity in the pericolic area. 

Currently, the TNM staging system is widely accepted for tumor staging globally, and also represents the main staging system in our country. The 8th revision is considered to be a major turning point in the evolution of cancer staging [20]. GCLN are considered non-regional, being classified as metastases (M1a) by TNM tumor staging, and as stage IVA by AJCC (American Joint Committee of Cancer) classification [20,21]. If these LN were considered as regional, based on the abovementioned anatomical and embryological considerations, as suggested in this paper, the staging would change to N in the TNM tumor staging, and at most in stage IIIC in the AJCC classification. Moreover, the surgical techniques should involve the excision of nodal stations 204, 206 and 214v as standard procedure during CME for proximal TCC.

Excision of GCLN in tumors near the hepatic flexure is currently under debate, and the most widely accepted indication is in case of suspicions of tumor involvement [9,22]. In our opinion, it is difficult to accurately assess clinically (CT/MRI) and even intraoperatively the tumor involvement of these LNs. Indocyanine green mapping of the LN for selected cases may be considered, but there are no comprehensive multicenter studies to prove the effectiveness of this method and the potential for recurrence over time [23]. In our study, we found no correlation between the number of GCLNs and size of GCL excised and their tumor involvement. However, we found that GCLN involvement is more likely to occur in tumors that grow through the muscularis propria (>T2). 

Published data showed that the incidence of GCLN tumor involvement in patients with TCC, including the hepatic flexure, varies between 0.7–22% [24], and seems to be associated with an aggressive disease and a poor prognosis. In this sense, GCLN metastasis was identified as an independent prognostic factor for patients with TCC [25]. The question that rises is whether all patients with TCC should have an extended lymphadenectomy, or the identification of preoperative risk factors of GCLN involvement could guide indication for extended lymphadenectomy [9,14,24]. From the oncologist’s point of view, a completely resected TNM stage IV benefits from the same adjuvant chemotherapy as a locally advanced TNM stage III, which is up to 6 months of perioperative treatment [26]. 

Over the years, it has been thought that performing radical oncological surgery can increase morbidity and mortality, but studies have proved otherwise [27]. Particularly for GCLN dissection, Huang et al. found that it may increase the incidence of gastroparesis [19]. In our study group, we recorded a low morbidity rate (7%) and no mortality. Most of our patients (74.4%) did not experience any complications. The most severe complication class in the Dindo–Clavien classification was IIIA, which occurred in three patients (7%). Patients in our study had no complications that required reoperation. Therefore, extensive dissection of GCLNs proved to be safe, similar to other studies findings [18]. The multivariate logistic analysis in the present study showed that a pT4 stage (*p* = 0.016) and lymphatic invasion (*p* < 0.001) were independently associated with GCLN metastases, similar to other studies findings [19]. The laparoscopic approach is preferred due to better visibility during GCL dissection, especially in the superficial pancreatic LN, preventing bleeding from the Anterior and superior pancreaticoduodenal veins and preventing pancreatic injury during GCL resection which led to the lack of postoperative complications and a shorter duration of postoperative hospitalization.

This study is a step in improving colon cancer surgery, whether we are individualizing or standardizing a particular surgical procedure. However, experienced teams in oncological surgery are required to perform this type of surgery.

Performing lymphadenectomy in GCLN can be associated with a lower risk of recurrence, but to demonstrate this, further studies on larger cohorts are needed. Even though, at this stage of research, clinical evidence does not suffice to support standardization, this hypothesis was proven feasible by our study and remains to be validated by future research. The main challenge that remains regarding GCLN excision in proximal TCC is to establish whether it warrants an improved overall survival.

Other limitations of this study are related to the limited number of cases. Further studies are needed to validate our findings.

## 5. Conclusions

Lymph node dissection of the gastrocolic ligament in patients with advanced proximal transverse colon cancer may improve the oncological outcome in T3/T4 tumors, and therefore standardization could be feasible.

## Figures and Tables

**Figure 1 medicina-58-00596-f001:**
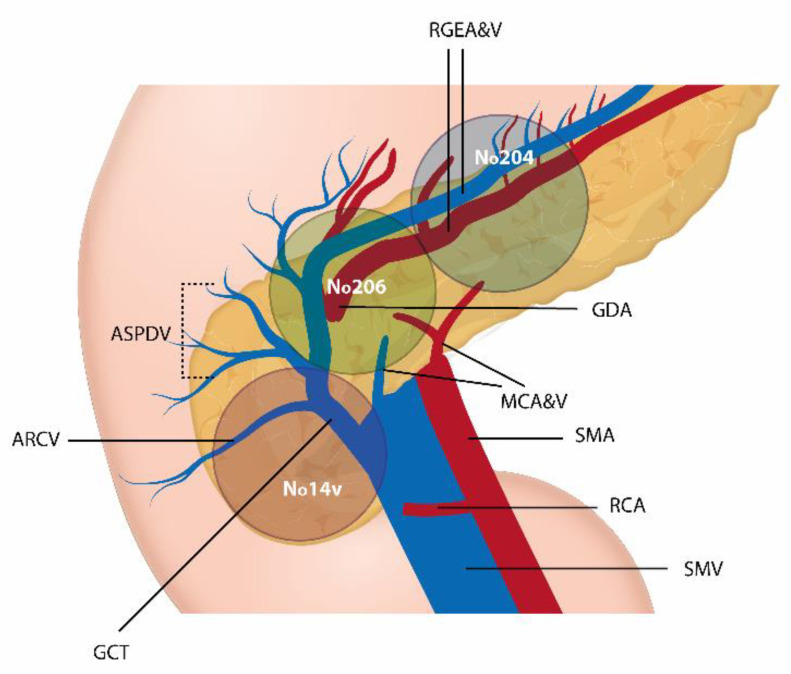
GCLN placement in relation to the mesenteric vessels, pancreas and stomach: gastroepiploic LN (No. 204), infrapyloric LN (No. 206) and superficial pancreatic LN (No. 214v). RGEA&V, Right Gastroepiploic Artery and Vein; ASPDV, Anterior Superior Pancreaticoduodenal Vein; ARCV, Accessory Right Colic Vein; GCT, Gastrocolic Trunk; GDA, Gastroduodenal Artery; MCA&V, Middle Colic Artery and Vein; SMA, Superior Mesenteric Artery; RCA, Right Colic Artery; SMV, Superior Mesenteric Vein.

**Figure 2 medicina-58-00596-f002:**
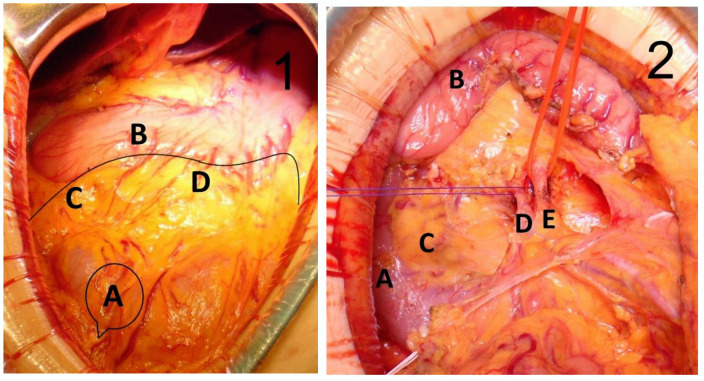
Intraoperative aspects. (**1**) Resection line of the GCL: A—tumor location in proximal transverse colon; B—stomach; C—No 206 infrapyloric LN; D—No 204 gastroepiploic LN. (**2**) A—duodenum; B—stomach; C—pancreas; D—middle colic vein; E—middle colic artery. (**3**) A—duodenum; B—stomach; C—pancreas; D—superior mesenteric vein. (**4**) Fresh specimen with GCL. A—tumor location; B—infrapyloric LN marked in blue line; C—gastroepiploic vessels and lymph nodes marked by blue line.

**Figure 3 medicina-58-00596-f003:**
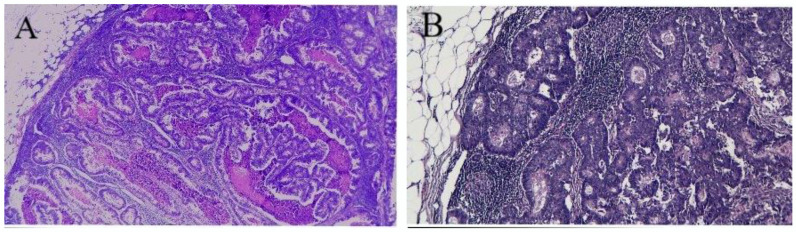
Pathological examination of LN. (**A**) HE, 4x: metastasis of colorectal adenocarcinoma in lymph node—there is epithelial neoplastic proliferation with cribriform pattern and comedonecrosis. (**B**) HE, 4x: cribriforming neoplastic glands delimited by atypical cylindrical epithelium, with comedonecrosis.

**Figure 4 medicina-58-00596-f004:**
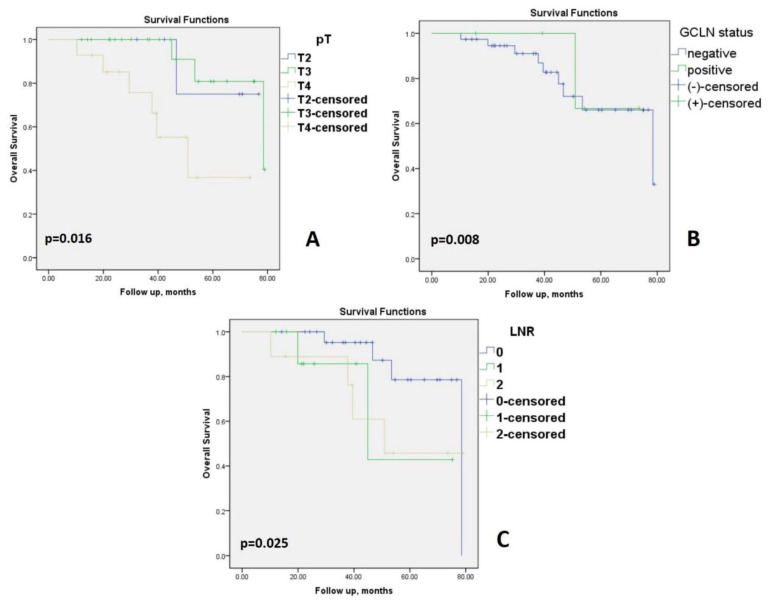
Kaplan–Meier overall survival analysis in patients with proximal transverse colon cancer. (**A**) Overall survival curves by stage pT2/3/4 (pathological tumor). (**B**) Kaplan–Meier survival curve by GCLN status. (**C**) Survival curves in different lymph node ratio groups; LNR = lymph node ratio: 0 = LNR < 0.05; 1 = LNR ranges from 0.05 to 0.20; 2 = LNR > 0.20.

**Table 1 medicina-58-00596-t001:** Patient demographics, clinical characteristics and surgical procedures.

Variables	Patients (n = 43)	Percentage (%)
Age ** (y)	65.09 ± 12.63 (35–86)	
Gender Male Female	18 25	41.9 58.1
BMI * Underweight Normal Overweight Obesity I Obesity II Obesity III	26.3 (17.8–43) 1 15 16 9 1 1	2.3 34.9 37.2 20.9 2.3 2.3
ACCI Mean ± SD 0–1 2–3 4–5 ≥6	4.95 ± 1.78 0 7 22 14	0 16.3 51.2 32.6
Abdominal operation history	4	9.3
Surgical approach Laparoscopic Open	13 30	30.2 69.8
Operation time (min) **	193.14 ± 22.15 (150–240)	
Estimate blood loss (mL) **	114.19 ± 35.87 (0–210)	
Postoperative hospital stay *	6 (5–11)	
Follow-up * (m)	40.77 (10.27–78.93)	

GCLN—gastrocolic ligament lymph node; y—years; BMI—Body Mass Index; ACCI—Age-adjusted Charlson Comorbidity Index; SD—Standard Deviation; m—months. With percentages in parentheses unless indicated otherwise, * Values are median (range), ** Values are mean (standard deviation) (range).

**Table 2 medicina-58-00596-t002:** Evaluation of postoperative complications according to the Clavien–Dindo classification.

Variables	Patients (n)	Percentage (%)
Complications	11	25.6
Reoperation	0	0
Clavien–Dindo classification		
Grade I	2	4.7
Grade II	8	18.6
Grade IIIA	3	7
Grade IIIB	0	0
Grade IVA	0	0
Grade IVB	0	0
Grade V	0	0
Most severe complication		
Grade I	1	2.3
Grade II	7	16.3
Grade III	3	7
Grade ≥IV	0	0

**Table 3 medicina-58-00596-t003:** Histopathological findings.

Variables		Patients (n = 43)	Percentage (%)
pTstage	T2	6	14
	T3	23	53.5
	T4	14	32.6
pN stage	N0	21	48.8
	N1	11	25.6
	N2	11	25.6
GCLN	Involvement	5	11.6
	No. 204 No. 204 + 206 No. 204 + 214v	3 1 1	7 2.3 2.3
LNR	<0.05	25	58.2
	≥0.05 to <0.20 ≥0.20	9 9	20.9 20.9
pM stage	M0	38	88.4
	M1a	5	11.6
Microscopical type	Adenocarcinoma Other	37 6	86 14
Tumor grade	Low grade High grade	34 9	79.1 20.9
Invasion	Venous	22	51.2
	Lymphatic	21	48.8
	Perineural	14	32.6
Stage	I	6	14
	II	15	34.9
	III	17	39.5
	IV	5	11.6
Adjuvant chemotherapy		22	51.2

pT, pathological tumor; pN, pathological node; LN, lymph node; GCLN, gastrocolic ligament lymph node; LNR, lymph node ratio; pM, pathological metastasis; With percentages in parentheses unless indicated otherwise, Values are median (range).

**Table 4 medicina-58-00596-t004:** Univariate and multivariate analysis for GCLN metastases.

Variables		Univariate Analysis			Multivariate Analysis	
		GCLN (−) n = 38	GCLN (+) n = 5	*p* Value	OR [95%CI]	*p* Value
Gender	Male Female	17 (44.7) 21 (55.3)	1 (20) 4 (80)	0.066		
Depth of tumor invasion	T3 T4	22 (57.9) 10 (26.3)	1 (20) 4 (80)	*0.019*	1.360 [1.075–3.067]	*0.016*
Tumor grade	Low grade High Grade	31 (81.6) 7 (18.4)	3 (60) 2 (40)	0.072	2.030 [0.820–5.051]	0.092
Lymphatic invasion	Present Absent	22 (57.9) 16 (42.1)	0 5 (100)	*0.011*	0.421 [0.290–0.611]	*0.001*
Venous invasion	Present Absent	21 (55.3) 17 (44.7)	0 5 (100)	*0.024*	0.447 [0.314–0.637]	0.144
Perineural invasion	Present Absent	29 (76.3) 9 (23.7)	0 5 (100)	*0.008*	0.237 [0.134–0.419]	0.250

Values represent numbers of patients (percentage), unless indicated otherwise. Values in italics indicate statistical significance (*p* < 0.050). GCLN—gastrocolic ligament lymph node; OR—odds ratio; CI—confidence interval; T—pathological tumor; pN—pathological node.

**Table 5 medicina-58-00596-t005:** Comparison of clinical and operative outcomes between laparoscopic and open surgery for T3 proximal transverse colon cancer.

Variables	Open Surgery (n = 16)	Laparoscopic Surgery (n = 7)	*p*-Value
Age * (y)	68.81 ± 10.66 (35–78)	53.29 ± 10.84 (36–67)	0.193
Gender Male Female	3 (18.8) 13 (81.3)	5 (71.4) 2 (28.6)	0.015
BMI *	26.03 ± 5.79 (17.8–38.1)	27.01 ± 4.06 (21.6–32.9)	0.481
ACCI 0–1 2–3 4–5 ≥6	0 1 (6.3) 9 (56.2) 6 (37.5)	0 3 (42.9) 4 (57.1) 0	0.163
Abdominal operation history	4 (25)	0	0.146
Operation time (min) *	181.56 ± 11.51 (160–210)	220 ± 20 (200–240)	0.007
Clavien–Dindo classification Grade I Grade II Grade IIIA ≥Grade IIIB	1 (6.3) 6 (37.5) 2 (12.5) 0	0 0 0 0	
Postoperative hospital stay *	6.44 ± 1.15 (5–10)	5.29 ± 0.49 (5–6)	0.026

Y—years; BMI—Body Mass Index; ACCI—Age-adjusted Charlson Comorbidity Index. With percentages in parentheses unless indicated otherwise, * Values are mean (standard deviation) (range).

## Data Availability

The data presented in this study are available on request from the authors.

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
