# Peer review of "Extended Lymphadenectomy for Proximal Transverse Colon Cancer: Is There a Place for Standardization?"

_medicina, 2022, doi:10.3390/medicina58050596_

Round 1

Reviewer 1 Report

Popescu and collegues designed a retrospective study to assess, in 43 patients with proximal transverse colon cancer tumors, the involvement of the gastrocolic ligament lymph nodes  (GCLN) (i.e. stations n° 204, 206, 214v) and to determine the applicability of GCLN dissection as the standard approach for these tumors. The authors found an incidence of 11.5% of tumor involvement of GCLN, which appeared associated with pT4 and lymphatic invasion.   

A number of issues deserve consideration.

In the Methods section (page 3) the authors state that patients with T2 tumors were selected for a laparoscopic approach versus those with T3-4 who underwent an open resection. First, it would be better to state that cT2 were selected for laparoscopy. Second, it is extremely difficult to differentiate between cT2 and cT3 tumors based on preoperative images. As a consequence, only 6 patients were ultimately pT2 while 13 underwent a laparoscopic procedure. A comment should be added and, in particular to justify exclusion from laparoscopy of cT3 patients.

In Table 1 Obesity I, II, III should be defined. Similarly, in Table 2 there are confusing data. It is not clear the difference between the Clavien-Dindo classification and the Most severe complication class. In these two parts numbers are not consistent. Please clarify.

In the results section (page 7 line 183), the authors report the “average” number of involved LN. It should be preferable using mean or median. The authors also report (table 3) the LNR (LN ratio) and the LODDS (log odds of positive LNs). They propose a stratification of the LODDS in three classe using 1.36 and 0.53 as cut offs: how did they define these cut-offs. In addition, while the LODDS is quite difficult to understand they did not use it for any comparison. To the reviewers opinion it seems just redundant.

In the Methods section, the authors state that “Independent risk factors for GCLN tumor involvement were determined using uni- and multivariate binary logistic regression models using long-term survival as the dependent variable, and patient and tumor characteristics as covariates”. However in the results section they simply did a multivariable binary logistic regression using GCLN+ as dependent variable and identified 2 factors independently predicting the risk of GCL lymph nodes metastases: pT4 tumors and lymphatic invasion. However: 1) in spite of a p-value of 0.016 the pT4 has a 95% CI of 0.753-3.067 which includes 1 thus making this association questionable; 2) the presence of lymphatic invasion appears as a protective effect with a OR of 0.421 (95%CI 0.290-0.611). These aspects should be commented.

One of the secondary end points was performing a comparison between the open and laparoscopic approaches. For this the authors only compared patients who underwent resection for T3 tumors. In the reviewer perspective this analysis is not clear. The authors “selected” for the laparoscopic approach patients with T2 tumors and after they excluded them from the analysis. Considering the small numbers, It would be more appropriate to use all patients. Nevertheless, from such comparison no novel data can be expected since larger RTCs have been performed comparing open versus laparoscopic colorectal resections.   

When analyzing the survival of patients (page 10, line 222) the authors affirm: OS in patients with positive GCLN was lower compared to the negative ones (p= 0.008). However, the survival curve (figure 4B green line) is always above the blue line crossing it at least after 45 months. In addition, comparing survival between two groups with one of only 5 patients appears at least meaningful.

In the discussion section the authors should highlight the relevance of their findings in the context of the available literature and whether their data might be or not sufficient for calling GCLN dissection as the standard approach for these tumors (as they stated in the instruction page 2 line 57-58 –aim of the study)     

Author Response

We thank the Editor and the Reviewers for their efforts. Thank you very much for your effort. We complied to the comments as follows:

                  1. In the Methods section (page 3) the authors state that patients with T2 tumors were selected for a laparoscopic approach versus those with T3-4 who underwent an open resection. First, it would be better to state that cT2 were selected for laparoscopy. Second, it is extremely difficult to differentiate between cT2 and cT3 tumors based on preoperative images. As a consequence, only 6 patients were ultimately pT2 while 13 underwent a laparoscopic procedure. A comment should be added and, in particular to justify exclusion from laparoscopy of cT3 patients.

                  Response: T4 tumors benefited from open surgery, T3 tumors usually underwent open surgery (but not exclusively), while T2 underwent laparoscopic surgery almost exclusively. Comorbidities were also considered when deciding the approach. Therefore, T3 tumors benefited from both open and laparoscopic interventions, allowing the comparison. In this sense, we added to the manuscript “Only T3 tumors were considered for this comparative analysis, as T4 tumors were exclusively operated by open approach, while T2 tumors were almost always operated using laparoscopy.” (rows 236-238)

                  2. In Table 1 Obesity I, II, III should be defined. Similarly, in Table 2 there are confusing data. It is not clear the difference between the Clavien-Dindo classification and the Most severe complication class. In these two parts numbers are not consistent. Please clarify.

                  Response: we have clarified by modifying Table 2 accordingly.

                  3. In the results section (page 7 line 183), the authors report the “average” number of involved LN. It should be preferable using mean or median.

                  Response: we have modified the text accordingly.

                  4. The authors also report (table 3) the LNR (LN ratio) and the LODDS (log odds of positive LNs). They propose a stratification of the LODDS in three class using 1.36 and 0.53 as cut offs: how did they define these cut-offs. In addition, while the LODDS is quite difficult to understand they did not use it for any comparison. To the reviewers opinion it seems just redundant.

                  Response: we have eliminated LODDS from the paper.

                  5. In the Methods section, the authors state that “Independent risk factors for GCLN tumor involvement were determined using uni- and multivariate binary logistic regression models using long-term survival as the dependent variable, and patient and tumor characteristics as covariates”. However in the results section they simply did a multivariable binary logistic regression using GCLN+ as dependent variable and identified 2 factors independently predicting the risk of GCL lymph nodes metastases: pT4 tumors and lymphatic invasion. However: 1) in spite of a p-value of 0.016 the pT4 has a 95% CI of 0.753-3.067 which includes 1 thus making this association questionable; 2) the presence of lymphatic invasion appears as a protective effect with a OR of 0.421 (95%CI 0.290-0.611). These aspects should be commented.

                  Response: The statistical methods were modified according to the reviewer’s indication. We have corrected the transcription errors in Table 4.

                  6. One of the secondary end points was performing a comparison between the open and laparoscopic approaches. For this the authors only compared patients who underwent resection for T3 tumors. In the reviewer perspective this analysis is not clear. The authors “selected” for the laparoscopic approach patients with T2 tumors and after they excluded them from the analysis. Considering the small numbers, It would be more appropriate to use all patients. Nevertheless, from such comparison no novel data can be expected since larger RTCs have been performed comparing open versus laparoscopic colorectal resections.  

                  Response: T4 tumors benefited from open surgery, T3 tumors usually underwent open surgery (but not exclusively), while T2 underwent laparoscopic surgery almost exclusively. Comorbidities were also considered when deciding the approach. Therefore, T3 tumors benefited from both open and laparoscopic interventions, allowing the comparison. Clearly, further studies are needed to validate the current paper. In this sense, we added to the manuscript “Only T3 tumors were considered for this comparative analysis, as T4 tumors were exclusively operated by open approach, while T2 tumors were almost always operated using laparoscopy.” (rows 236-238)

                  7. When analyzing the survival of patients (page 10, line 222) the authors affirm: OS in patients with positive GCLN was lower compared to the negative ones (p= 0.008). However, the survival curve (figure 4B green line) is always above the blue line crossing it at least after 45 months. In addition, comparing survival between two groups with one of only 5 patients appears at least meaningful.

                  Response: We have modified the text: “OS in patients with positive GCLN was similar compared to the negative ones (p= 0.008); note that the GCLN positive group had only 5 patients (Fig. 4B)” (rows 253-258). We also eliminated the analysis on diabetes, as it is not relevant for the study.

                  8. In the discussion section the authors should highlight the relevance of their findings in the context of the available literature and whether their data might be or not sufficient for calling GCLN dissection as the standard approach for these tumors (as they stated in the instruction page 2 line 57-58 –aim of the study)    

                  Response: We modified the manuscript accordingly:

                  “Even though, at this stage of research, clinical evidence does not suffice to support standardization, this hypothesis was proven feasible by our study and remains to be validated by future research.”. (rows 349-352)

                  The Conclusion was also modified: “Lymph node dissection of the gastrocolic ligament in patients with advanced proximal transverse colon cancer may improve the oncological outcome, and therefore standardization could be feasible.” (rows 358-359)

Reviewer 2 Report

From clinical point of view the manuscript is very interesting. Hovewer, The autors declare only 43 patients who were evaluated so the conclusion "Lymph node dissection of the gastrocolic ligament in patients with advanced proximal transverse colon cancer should become standard in T3/T4 tumors." in my opinion is not enougt from the scientific point of view. Among all investigated patients those with T2 were 6, T3 n= 23 and
T4
n=14. Did authors calculate the minimum n of patients from statistical point of view?

Author Response

We thank the Editor and the Reviewers for their efforts. Thank you very much for your effort. We complied to the comments as follows:

                  1. Hovewer, the autors declare only 43 patients who were evaluated so the conclusion "Lymph node dissection of the gastrocolic ligament in patients with advanced proximal transverse colon cancer should become standard in T3/T4 tumors." in my opinion is not enougt from the scientific point of view.

                  Response: We agree with the reviewer, and we modified the manuscript accordingly:

                  “Even though, at this stage of research, clinical evidence does not suffice to support standardization, this hypothesis was proven feasible by our study and remains to be validated by future research.”. (rows 349-352)

                  The Conclusion was also modified: “Lymph node dissection of the gastrocolic ligament in patients with advanced proximal transverse colon cancer may improve the oncological outcome, and therefore standardization could be feasible.” (rows 358-359)

                  2. Among all investigated patients those with T2 were 6, T3 n= 23 and T4 n=14. Did authors calculate the minimum n of patients from statistical point of view?

                  Response: We were aware that the number of case is insufficient to draw a statistical conclusion, and therefore no minimum number of patients was considered from this point of view.

Reviewer 3 Report

Overall, a good quality manuscript, though the number of patients included in the study is small (noted as potential weakness or limitation of the publication by the authors themselves).

More data from the papers related to the topic could be included:

Huang S, Wang X, Deng Y, Jiang W, Huang Y, Chi P. Gastrocolic Ligament Lymph Node Dissection for Transverse Colon and Hepatic Flexure Colon Cancer: Risk of Nodal Metastases and Complications in a Large-Volume Center. J Gastrointest Surg. 2020 Nov;24(11):2658-2660. doi: 10.1007/s11605-020-04705-4. Epub 2020 Jul 14. PMID: 32666497.

Current conclusion that the lymph node dissection of the gastrocolic ligament in patients with advanced proximal transverse colon cancer should become standard in T3/T4 tumors is a little bit too far fetched based on such a small study. I would argue that the conclusion should rely on the findings, i.e. gastrocolic ligament node involvement is observed in patients with  proximal transverse colon T3/T4 tumors and the excision of the GCL is not associated with increased perioperative morbidity. Then discussion statement that data suggests this should be a standard procedure for advanced tumors but current findings merit validation in larger patient series.

Author Response

We thank the Editor and the Reviewers for their efforts. Thank you very much for your effort. We complied to the comments as follows:

                  1. More data from the papers related to the topic could be included:

Huang S, Wang X, Deng Y, Jiang W, Huang Y, Chi P. Gastrocolic Ligament Lymph Node Dissection for Transverse Colon and Hepatic Flexure Colon Cancer: Risk of Nodal Metastases and Complications in a Large-Volume Center. J Gastrointest Surg. 2020 Nov;24(11):2658-2660. doi: 10.1007/s11605-020-04705-4. Epub 2020 Jul 14. PMID: 32666497.

                  Response: the suggested paper was included in the text and references:

“Particularly for GCLN dissection, Huang et al found that it may increase the incidence of gastroparesis (19).” (Rows 331-333)

“The multivariate logistic analysis in the present study showed that a pT4 stage (p = 0.016) and lymphatic invasion (p<0.001) were independently associated with GCLN metastases, similar to other studies findings (19).” (rows 338-340)

                  2. Current conclusion that the lymph node dissection of the gastrocolic ligament in patients with advanced proximal transverse colon cancer should become standard in T3/T4 tumors is a little bit too far fetched based on such a small study. I would argue that the conclusion should rely on the findings, i.e. gastrocolic ligament node involvement is observed in patients with  proximal transverse colon T3/T4 tumors and the excision of the GCL is not associated with increased perioperative morbidity. Then discussion statement that data suggests this should be a standard procedure for advanced tumors but current findings merit validation in larger patient series.

                  Response: We agree with the reviewer, and we modified the manuscript accordingly:

                  “Even though, at this stage of research, clinical evidence does not suffice to support stand-ardization, this hypothesis was proven feasible by our study and remains to be validated by future research.”. (rows 349-352)

                  The Conclusion was also modified: “Lymph node dissection of the gastrocolic ligament in patients with advanced proximal transverse colon cancer may improve the oncological outcome, and therefore standardization could be feasible.” (rows 358-359)

Round 2

Reviewer 1 Report

The authors addressed all the reviewers' comments.